# On the Measurement of Ocean Near-Surface Current from a Moving Buoy

Carlos F. Herrera-Vázquez [1,*] , Nicolas Rascle [1,2] , Francisco J. Ocampo-Torres [3] , Pedro Osuna [1] and Héctor García-Nava [4]

1   Departamento de Oceanografía Física, Centro de Investigación Científica y Educación Superior de Ensenada (CICESE), Ensenada 22860, Mexico; nicolas.rascle@ifremer.fr (N.R.); osunac@cicese.mx (P.O.)
2   Brest University, Centre National de la Recherche Scientifique (CNRS), Institut Français de Recherche pour L'exploitation de la mer (Ifremer), Institut de Recherche pour le Développement (IRD), Laboratoire d'Océanographie Physique et Spatiale (LOPS), IUEM, F29280 Plouzané, France
3   Centro Mexicano de Innovación en Energía del Océano (CEMIE-Océano), C.U., Coyoacan, Ciudad de México 04510, Mexico; pocampotorres@gmail.com
4   Instituto de Investigaciones Oceanológicas, Universidad Autónoma de Baja California (UABC), Ensenada 22860, Mexico; hector.gnava@uabc.edu.mx
*   Correspondence: cherrera@cicese.edu.mx or cf.herrera.vazquez@gmail.com

**Abstract:** This paper studies the error that occurs when measuring surface currents with a current meter mounted on a buoy or a mooring line whose horizontal and vertical motions respond to the presence of waves. The error is defined with respect to an Eulerian reference measurement where the sensor does not move. First, we present the subject with a theoretical analysis in the case of a monochromatic wave. That idealized model allows us to study particular sensor or mooring line motions. Second, a realistic numerical model is implemented to reconstruct the current field with a high resolution near the surface. Wave orbital velocities are generated with a random phase model. An Ekman-type current, uniform in the horizontal but with a vertical shear, is also incorporated. The results indicate that the error in the current measurement is highly dependent on the sensor motion induced by waves. The error magnitude is proportional to the wave momentum or Stokes drift and depends on the wave development state and the wind-generated current's magnitude. The error obtained in the current measurement is analyzed by considering that the buoy only responds to low-frequency waves up to a maximum frequency. That maximum frequency is referenced concerning the peak frequency of the third moment of the spectrum (i.e., the Stokes drift spectrum). It allows us to classify the current time average into three ranges with respect to the maximum frequency: (1) Eulerian average, (2) wave-following average, and (3) intermediate case of undulating average where results cannot be generalized. The measurement error is most important in the region above the wave troughs. However, the error is also considerable in the region confined below the wave troughs and down to the Stokes drift e-folding depth. The error is particularly relevant in conditions of developed and energetic waves ($Hs > 3$ m), where the surface Stokes drift can reach values above 0.1 m/s. It should be noted that measurement error can exceed the value of the Stokes drift at the sensor depth for certain mooring line motions. Those results should help better interpret in situ near-surface current measurements obtained from various devices.

**Keywords:** near-surface vertical shear; stokes drift; Ekman current; Eulerian current; Quasi-Eulerian current; wave bias

## 1. Introduction

Observations of ocean surface characteristics are required to understand the dynamical processes occurring within the marine boundary layer, including surface velocities and waves, essential to transferring properties between the ocean and the atmosphere. In situ measurements have been carried out to meet these needs with different types of buoys [1,2]

equipped with appropriate sensors to study the characteristics of the ocean surface layer. However, velocity measurements made by current meters installed on buoys are commonly analyzed in a fixed framework without considering the wave-induced motion of the buoys [3,4]. In the case of velocity measurements, the presence of waves induces an intense vertical shear very close to the surface, which, together with the displacement of the sensor, introduces errors that must be rigorously analyzed [3].

Pollard (1973) [5] analytically identified that when a current meter moves vertically and perfectly follows the sea surface elevation, this movement induces an error in the average time velocity proportional to half of the Stokes drift. He called that error the wave–bias in the current measurement due to the presence of swell. Collar et al. (1983) [6] validated the previous results with laboratory data and analyzed that bias in the measurement when current meters are mounted on buoys of finite dimension. Santala and Terray (1992) [7] proposed a technique to perform unbiased measurements of the vertical current shear using a wave-following current meter. They conducted the study using a numerical model and field observations, where they identified the need to measure the current meter position. While they investigated the extreme cases of buoy motions—both a fixed and a wave-following buoy—error analysis has not been carried out for intermediate cases, where the buoy motion responds only to specific wave scales.

The scientific community has studied the correction of current velocities measured with a sensor whose movement has responded to the presence of waves for a few decades. Nonetheless, this research has been limited for many years by the need for sensors with good accuracy and high sampling frequency close to the surface. In the last decade, the subject has regained interest with the emergence of autonomous underwater vehicles and the development of new technologies. Amador et al. (2017) [8] identified that the current measurement by autonomous vehicles is affected, among other factors, by the movement of these vehicles, the presence of waves, and mean currents. On the other hand, there is a growing need to validate and compare current measurements obtained at the surface, close to the surface, and at the subsurface by different techniques. Those techniques include remote measurements from an airplane or satellite-mounted radars [9], X-band radars [10,11], and comparisons with measurements obtained by ADCPs moored and mounted on buoys [4].

In this paper, we study numerically the current measurement error when a current meter is mounted on a buoy whose horizontal and vertical displacements are induced by the presence of waves. Section 2 has been separated into subsections to overview the problem comprehensively. First, an overview of the framework used to describe measurements made from moving buoys is presented. Next, the analytical development of measurements obtained by a current meter whose displacements perfectly follow the sea surface in the presence of wave orbital velocity (wave orbital bias) or an average current with a vertical shear profile (average current bias) is shown. Additionally, the implementation of a model that reproduces the velocity field with a high resolution in the vertical (in the first meters of the surface) from a wave spectrum is described. This allows us to study the measurement of currents by a mobile buoy in realistic conditions. Section 3 analyzes the currents of various sea states from an Eulerian and undulating framework. Section 4 identifies the error in the measurement of surface currents considering various ranges of buoy motion.

## 2. Materials and Methods

### 2.1. Eulerian, Quasi-Eulerian, and Undulating Mean Current

In general, one distinguishes two ways to measure fluid velocities, the Eulerian and the Lagrangian descriptions (e.g., [12]). Measurements made under the Eulerian description are taken at fixed locations, whereas in the Lagrangian description, the velocity is measured by following a particular volume of fluid.

When velocities are measured close to the free surface in the presence of waves (Figure 1a), the analysis is complicated by two important aspects. (1) Current meters used to make the measurements are commonly deployed on buoys, which move due to waves.

Hence, the measurements cannot be considered Eulerian, nor can they be considered Lagrangian. (2) Waves produce orbital velocities $\mathbf{u}^w(\mathbf{x}, t) = (u^w, v^w, w^w)$ which do not necessarily average out in time and will therefore give a contribution to the measurements. The magnitude of these orbital velocities is proportional to the wave slope $ak$ (where $a$ is the wave amplitude and $k$ the wave number).

More specifically, in the Eulerian description, the mean velocity profile (Eulerian-mean) is confined to $-h < z \leq \eta$ (where $h$ is the ocean bottom and $\eta$ is the instantaneous free surface), whereas in the Lagrangian description, the mean velocity profile (Lagrangian-mean) is restricted to $-h < z \leq 0$ (where 0 is the mean sea level, see Figure 1).

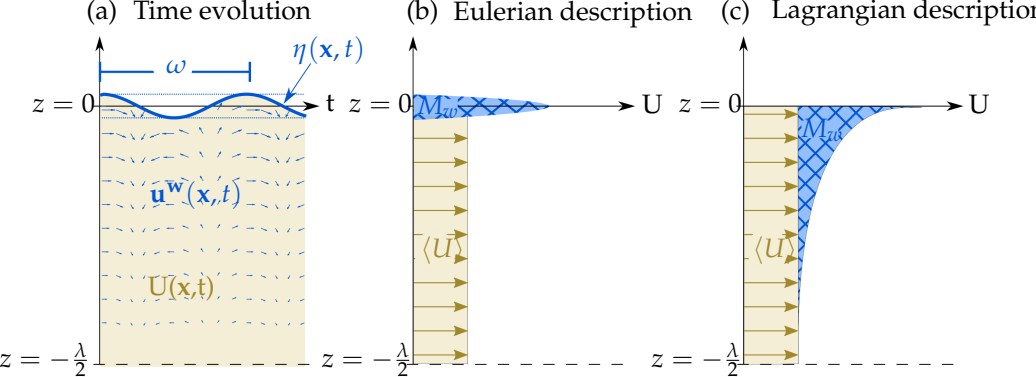

**Figure 1.** Schematic representation of the (**a**) field of orbital velocities $\mathbf{u}^w(\mathbf{x}, t)$ and permanent current $U(\mathbf{x}, t)$ in the presence of waves $\eta(\mathbf{x}, t)$, and their respective time averages $M_w$ and $\langle U \rangle$ under a (**b**) Eulerian and (**c**) Lagrangian description. The vertical axis in each figure is indicated up to a depth $z = -\lambda/2$ (with $\lambda$ the wavelength).

Let us focus first on wave orbital velocities. The mean of the wave orbital velocities corresponds to the wave momentum $M^w$ (or wave mass transport or Stokes transport), which has a different representation depending on the framework or description that is considered (Figure 1b,c). When an Eulerian mean is applied, the wave momentum $M^w$ occurs between the crests and troughs of the waves and is zero below these (Figure 1b). $M^w$ is distributed vertically following a parabolic profile (also shown in Figure 2). The physical reason is the absence and presence of water during the wave passing. Therefore, surface-current measurements are commonly analyzed only below the wave troughs. When a Lagrangian mean is performed, the wave momentum is distributed vertically, with the highest values at the surface $z = 0$, as shown in Figure 1c.

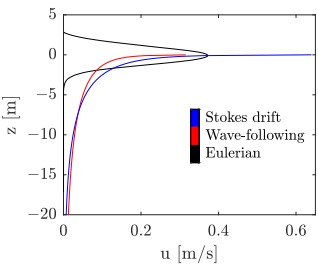

**Figure 2.** Schematic representation of the Eulerian mean (black) and an undulating mean (red) considering only vertical displacements (the wave-following mean). The Stokes drift is also shown (blue). Here, the wave is considered monochromatic.

Let us now consider the presence of a permanent current (both steady and horizontally homogeneous) with a vertical shear of the type $\mathbf{U} = (u(z), v = 0, w = 0)$. The Eulerian and Lagrangian averages will be equivalent if there are no waves. However, in the presence of waves, the average of the measurement of the permanent current with a vertical shear will be modified due to the disturbance of the free surface $\eta$.

If surface velocities are measured from a platform that remains fixed in space, the time average of these velocities will correspond to the Eulerian mean. However, if waves induce horizontal and vertical displacements, a portion of the wave momentum will be captured. The time average of these measurements corresponds neither to an Eulerian mean nor to a Lagrangian mean. Therefore, it will be referred to generally as an undulating mean. One particular case of the undulating mean is the wave-following mean.

Regardless of the averaging employed, it is helpful to express surface current measurements in an Eulerian framework to implement and validate numerical models. The generalized Lagrangian mean (GLM) concept was proposed by [12] to represent Lagrangian measurements in Eulerian numerical models. The GLM involves calculating the temporal average of a volume of fluid that is perturbed regarding the reference position in such a way that it yields

$$\overline{u}^L(\mathbf{x}, t) = \overline{u(\mathbf{x} + \delta\mathbf{x}, t)}. \tag{1}$$

According to Ardhuin et al. (2008) [13], the generalized Lagrangian average can be expressed as a quasi-Eulerian mean component modified by the presence of waves plus the wave pseudo-momentum (p), which corresponds to the Stokes drift $U_s$, resulting in the following expression,

$$\overline{u}^L(z) = u^{QE}(z) + U_s(z). \tag{2}$$

In the particular case in which the superficial velocities are measured following the trajectory of the particles of the fluid described by the linear wave theory, the average of these velocities will correspond to the Stokes drift.

Different average profiles of the horizontal component of the orbital velocity of waves are shown in Figure 2 to illustrate the averages introduced in this section. The black line corresponds to the Eulerian mean, the red line shows the average velocity field when a sensor moves vertically following the elevation of the free surface, and the blue line corresponds to the sensor moving horizontally and vertically following the trajectory of the orbital velocities, which is equivalent to the Stokes drift.

### 2.2. Theoretical Wave-Bias in Monochromatic Case

A horizontal current in the ocean $u(x, z, t)$ is considered, where the presence of waves modifies the upper boundary. For the sake of simplicity, we only consider one horizontal dimension $x$, ignoring the $y$ dimension whose contribution shall be added similarly. The measurement obtained by a current meter can be expressed as $u(X(t), Z(t), t)$ with $X(t), Z(t)$ the sensor position, and the mean current field will depend on the measurement framework, as well as the horizontal and vertical current distribution. In the general case, $X(t) = x_0 + \delta x(t)$ and $Z(t) = z_0 + \delta z(t)$. For an Eulerian measurement, the position is fixed in space, $X = x_0$ and $Z = z_0$. In the specific case of a wave-following buoy, $\delta x(t) = -a\sin(kx - \omega t)\exp kz_0$ and $\delta z(t) = a\cos(kx - \omega t)\exp kz_0$, where $a$ is the wave amplitude, $k$ the wave number and $\omega$ the angular frequency. As will be obtained below, in such case, a wave bias appears in the temporal average of the velocity [5].

The analytical development for calculating the wave bias is well-described in the literature. For a single vertical dimension $z$, it can be calculated by supposing a small perturbation $\delta z$ concerning the mean sensor position $z_0$ and using a Taylor series expansion of the wave orbital velocities. From linear wave theory, it is known that the horizontal wave orbital velocities are in phase with the wave surface elevation.

In the following section, a general analytical development will be presented for velocity fields and sensor displacements with unique characteristics, such as being in phase or quadrature. This general development encompasses different measurement configurations, including the presence of wave orbital velocities, mean surface current, and different types of sensor displacements.

### 2.2.1. Vertical Displacements

We start by considering only vertical sensor displacements, with no horizontal displacement. In the presence of a vertical displacement $\delta z$ around a mean position $z_0$, the disturbance in the measurement of a horizontal velocity field $u = u(x, Z(t), t)$ can be approximated by a Taylor series expansion as follows:

$$u(x, Z(t), t) = u(x, z_0 + \delta z, t) = \sum_{n=0}^{\infty} \frac{\delta z^n}{n!} \frac{\partial^n}{\partial z_0^{\,n}} u(x, z_0, t)$$

$$= u(x, z_0, t) + \delta z u'(x, z_0, t) + \frac{\delta z^2}{2} u''(x, z_0, t) + O(\delta z^3) \tag{3}$$

where prime values correspond to the partial derivatives of $u$ with respect to $z$. By applying a time average to Equation (3), we obtain:

$$\langle u(x, Z(t), t) \rangle_t = \langle u(x, z_0, t) \rangle_t + \langle \delta z u'(x, z_0, t) \rangle_t + \left\langle \frac{\delta z^2}{2} u''(x, z_0, t) \right\rangle_t + O\left(\delta z^3\right)_t \tag{4}$$

with the time average defined as

$$\langle \rangle_t = \frac{\omega}{2\pi} \int_0^{2\pi/\omega} () \, dt.$$

Next, we consider that the velocity field $u$ can be separated into a component that contains the horizontal and temporal variations and a component that contains the vertical variation,

$$u(x, Z(t), t) = F(x, t) G(z) \tag{5}$$

applying a time average to Equation (5), we obtain

$$\langle u(x, Z(t), t) \rangle_t = \langle F(x, t) G(z_0) \rangle_t + \langle \delta z F(x, t) G'(z_0) \rangle_t + \left\langle \frac{\delta z^2}{2} F(x, t) G''(z_0) \right\rangle_t + O\left(\delta z^3\right)$$

$$= G(z_0) \langle F(x, t) \rangle_t + G'(z_0) \langle \delta z F(x, t) \rangle_t + G''(z_0) \left\langle \frac{\delta z^2}{2} F(x, t) \right\rangle_t + O\left(\delta z^3\right) \tag{6}$$

From Equation (6), we can consider two particular cases based on whether the components containing horizontal and temporal variations are in phase (case A) or not (case B) with respect to $\delta z$.

Case A: Orbital Velocities (Surface Disturbance in Phase with Velocity Field)

If we consider that $F(x, t)$ is proportional to $\delta z$ and $\delta z$ corresponds to a periodic function (i.e., $F(x, t)$ is in phase with $\delta z$), such that $F(x, t) = \alpha \delta z$, where $\alpha$ is a constant of proportionality. In the case of wave orbital velocities, $\alpha = \omega$, and have dimensions $s^{-1}$, then we obtain

$$\langle u(x, Z(t), t) \rangle_t = G(z_0) \langle \alpha \delta z \rangle_t + G'(z_0) \langle \alpha \delta z^2 \rangle_t + G''(z_0) \left\langle \frac{\alpha \delta z^3}{2} \right\rangle_t + O\left(\delta z^4\right) \tag{7}$$

Due to the periodicity of $\delta z$, the terms in Equation (7) that include $(\delta_z^n)$ for odd $n$ will be zero when we perform the time average. Therefore,

$$\langle u(x, Z(t), t) \rangle_t = G'(z_0) \langle \alpha \delta z^2 \rangle_t + O\left(\delta z^4\right) \tag{8}$$

Keeping the terms to second order, we obtain

$$\langle u(x, Z(t), t) \rangle_t = G'(z_0) \langle \alpha \delta z^2 \rangle_t \tag{9}$$

If we consider that $\alpha$ does not depend on time and recall that $\delta z$ is a periodic function around 0,

$$\langle u(x, Z(t), t) \rangle_t = \alpha G'(z_0) Var(\delta z) \tag{10}$$

Case B: Permanent Current with Curvature (Disturbance Not in Phase with the Velocity Field)

If $F(x,t)$ varies much more slowly than the wave-related variations of the vertical disturbance $\delta z$, then $F(x,t)$ can be considered a permanent velocity field $F(x)$ and the Equation (6) reduces to

$$\langle u(x, Z(t), t)\rangle_t = F(x)G(z_0) + F(x)G'(z_0)\langle \delta z\rangle_t + F(x)G''(z_0)\left\langle \frac{\delta z^2}{2} \right\rangle_t + O\left(\delta z^3\right) \quad (11)$$

In addition, if $\delta z$ corresponds to a periodic function of small amplitude around an average value of 0, we obtain

$$\langle u(x, Z(t), t)\rangle_t = F(x)G(z_0) + F(x)G''(z_0)\left\langle \frac{\delta z^2}{2} \right\rangle_t + O\left(\delta z^3\right) \quad (12)$$

Considering the second-order approximation, we identify that the temporal average of a velocity field for the proposed hypotheses depends on whether it is in phase with the surface disturbance.

$$\langle u(x, Z(t), t)\rangle_t = F(x)G(z_0) + F(x)G''(z_0)\left\langle \frac{\delta z^2}{2} \right\rangle_t \quad (13)$$

$$\langle u(x, Z(t), t)\rangle_t = F(x)G(z_0) + F(x)G''(z_0)\frac{\mathrm{Var}(\delta z)}{2} \quad (14)$$

From (10) and (14), it is important to remember that a function's first and second derivatives are related to the vertical shear and curvature of the velocity profile, respectively.

### 2.2.2. Vertical and Horizontal Displacements

We now include horizontal sensor displacement $\delta x$ in addition to vertical displacement $\delta z$. The multivariable Taylor series expansion is given by the relationship [14]

$$u(\hat{x}_1, \cdots, \hat{x}_n) = u(x_1 + \delta x_1, \cdots, x_n + \delta x_n) = \sum_{j=0}^{\infty} \frac{1}{j!}\left[\sum_{k=1}^{n} \delta_{x_k}\frac{\delta}{\delta x_k}\right]^j f(x_1, \cdots, x_n) \quad (15)$$

where $\delta_{x_n} = \hat{x}_n - x_n$, and correspond to the different dimensions of a field u. Considering the velocity, these are given as field measurements by a current meter as $u = u(X(t), Z(t), t)$, and performing a Taylor expansion around $X(t), Z(t)$ such that $(\hat{x}_1, \hat{x}_2) = (x, z)$, and $(\delta x, \delta z) = (\hat{x} - x_0, \hat{z} - z_0)$, we obtain

$$
\begin{aligned}
u(X(t), Z(t), t) =& u(x_0 + \delta x, z_0 + \delta z, t) = \sum_{j=0}^{\infty}\frac{1}{j!}\left[\delta x\frac{\partial}{\partial x_0} + \delta z\frac{\partial}{\partial z_0}\right]^j u(x_0, z_0, t) \\
=& u(x_0, z_0, t) + \left[\delta x\frac{\partial}{\partial x_0} + \delta z\frac{\partial}{\partial z_0}\right]u(x_0, z_0, t) \\
&+ \frac{1}{2}\left[\delta x^2\frac{\partial^2}{\partial x_0^2} + 2\delta x\delta z\frac{\partial}{\partial x_0}\frac{\partial}{\partial z_0} + \delta z^2\frac{\partial^2}{\partial z_0^2}\right]u(x_0, z_0, t) + O(\delta x^3, \delta z^3)
\end{aligned}
\quad (16)
$$

$$
\begin{aligned}
u(X(t), Z(t), t) =& u(x_0 + \delta x, z_0 + \delta z, t) = \sum_{j=0}^{\infty}\frac{1}{j!}\left[\delta x\frac{\partial}{\partial x_0} + \delta z\frac{\partial}{\partial z_0}\right]^j u(x_0, z_0, t) \\
=& u(x_0, z_0, t) + \left[\delta x\frac{\partial}{\partial x_0} + \delta z\frac{\partial}{\partial z_0}\right]u(x_0, z_0, t) \\
&+ \frac{1}{2}\left[\delta x^2\frac{\partial^2}{\partial x_0^2} + 2\delta x\delta z\frac{\partial}{\partial x_0}\frac{\partial}{\partial z_0} + \delta z^2\frac{\partial^2}{\partial z_0^2}\right]u(x_0, z_0, t) + O(\delta x^3, \delta z^3)
\end{aligned}
\quad (17)
$$

If we consider $u(X(t), Z(t), t) = F(x, t)G(z)$, and use the notation $f_x(x) = \frac{\partial}{\partial x}f(x)$, we can substitute by:

$$
\begin{aligned}
u(X(t), Z(t), t) =& F(x_0, t)G(z_0) + \delta x F_{x_0}(x_0, t)G(z_0) + \delta z F(x_0, t)G_{z_0}(z_0) \\
&+ \frac{1}{2}\Big[\delta x^2 F_{x_0 x_0}(x_0, t)G(z_0) + 2\delta x \delta z F_{x_0}(x_0, t)G_{z_0}(z_0) + \delta z^2 F(x_0, t)G_{z_0 z_0}(z_0)\Big] + O(\delta x^3, \delta z^3).
\end{aligned}
\tag{18}
$$

Following the same reasoning as in the one-dimensional case of the previous section, we can now consider two particular hypotheses based on Equation (18).

Case A: Orbital Velocities (Perturbations In-Phase and Quadrature with the Velocities Field)

We first consider that $F(x, t) = \alpha_0 \delta z$, and $\delta x$ is in quadrature with $\delta z$ (phase shift of $\pi/2$), where $\delta z$ is a function of time and is represented as a wave-like function around $z_0$. Under these considerations, we have the following relationship of the $n$th derivatives of $F(x)$.

$$
\frac{\partial_n}{\partial x_0^n}F(x_0, t) = \begin{cases} \alpha_n \delta z & \text{for n even} \\ \beta_n \delta x & \text{for n odd} \end{cases}
$$

therefore

$$
\begin{aligned}
u(X(t), Z(t), t) =& \alpha_0 \delta z G(z_0) + \delta x \beta_1 \delta x G(z_0) + \delta z \alpha_0 \delta z G_{z_0}(z_0) \\
&+ \frac{1}{2}\Big[\delta x^2 \alpha_2 \delta z G(z_0) + 2\delta x \delta z \beta_1 \delta x G_{z_0}(z_0) + \delta z^2 \alpha_0 \delta z G_{z_0 z_0}(z_0)\Big] + O(\delta x^3, \delta z^3)
\end{aligned}
\tag{19}
$$

$$
\begin{aligned}
u(X(t), Z(t), t) =& \alpha_0 \delta z G(z_0) + \beta_1 \delta x^2 G(z_0) + \alpha_0 \delta z^2 G_{z_0}(z_0) \\
&+ \frac{1}{2}\Big[(\alpha_2 G(z_0) + 2\beta_1 G_{z_0}(z_0))\delta x^2 \delta z + \alpha_0 \delta z^3 G_{z_0 z_0}(z_0)\Big] + O(\delta x^3, \delta z^3)
\end{aligned}
\tag{20}
$$

Because $(\delta x, \delta z)$ correspond to periodic functions in quadrature, upon taking the time average of (20), the terms where $(\delta_x^m, \delta_z^n, \delta_x^m \delta_z^n)$ with odd $m, n$ or an odd combination of $m + n$ will be zero. Thus, we have

$$
\langle u(X(t), Z(t), t)\rangle_t = G(z_0)\langle \beta_1 \delta x^2 \rangle_t + G_{z_0}(z_0)\langle \alpha_0 \delta z^2 \rangle_t + O(\delta x^3, \delta z^3)
\tag{21}
$$

Considering only a second-order approximation in $\delta_z$, and recalling that $\delta z$ corresponds to a periodic function around 0, we have

$$
\langle u(X(t), Z(t), t)\rangle_t = G(z_0)\langle \beta_1 \delta z^2 \rangle_t + G_{z_0}(z_0)\langle \alpha_0 \delta z^2 \rangle_t
\tag{22}
$$

$$
\langle u(X(t), Z(t), t)\rangle_t = G(z_0)Var(\beta_1 \delta x) + G_{z_0}(z_0)Var(\alpha_0 \delta z)
\tag{23}
$$

When the velocity field is considered as the horizontal and vertical components of the orbital velocities and $\delta x, \delta z$ correspond to the trajectory followed by fluid particles, the expression obtained in (23) corresponds to the Stokes drift described by [13].

Case B: A Permanent Current with Curvature (Perturbations In-Phase or Out-of-Phase with the Velocity Field)

If $\delta z$ is in quadrature with $\delta x$, and if we consider that the velocity field does not vary much during the averaging time, we have $u(x, z, t) \approx u(x, z)$, and it cannot be expressed as a function of $\delta x$ and $\delta z$. Then, the Equation (18) reduces to

$$
\begin{aligned}
u(X(t), Z(t)) =& F(x_0)G(z_0) + \delta x F_{x_0}(x_0)G(z_0) + \delta z F(x_0)G_{z_0}(z_0) \\
&+ \frac{1}{2}\Big[\delta x^2 F_{x_0 x_0}(x_0)G(z_0) + 2\delta x \delta z F_{x_0}(x_0)G_{z_0}(z_0) + \delta z^2 F(x_0)G_{z_0 z_0}(z_0)\Big] + O(\delta x^3, \delta z^3).
\end{aligned}
\tag{24}
$$

after temporal averaging, we obtain

$$\langle u(X(t), Z(t)) \rangle_t = F(x_0)G(z_0) + \frac{1}{2}\left[ F_{x_0 x_0}(x_0)G(z_0)\langle \delta x^2 \rangle_t 2\langle \delta x \delta z \rangle_t F_{x_0}(x_0)G_{z_0}(z_0) + F(x_0)G_{z_0 z_0}(z_0)\langle \delta z^2 \rangle_t \right] \\ + O(\delta x^3, \delta z^3), \tag{25}$$

and recalling that terms $(\delta_x^n, \delta_z^n)$ will average to zero for odd n, we obtain

$$\langle u(X(t), Z(t)) \rangle_t = F(x_0)G(z_0) + F_{x_0 x_0}(x_0)G(z_0)\left\langle \frac{\delta x^2}{2} \right\rangle_t + F(x_0)G_{z_0 z_0}(z_0)\left\langle \frac{\delta z^2}{2} \right\rangle_t + O(\delta x^3, \delta z^3). \tag{26}$$

Up to second order in $\delta_z$, we can obtain from Equations (21) and (26):

$$\langle u(X(t), Z(t)) \rangle_t = F(x_0)G(z_0) + F_{x_0 x_0}(x_0)G(z_0)\left\langle \frac{\delta x^2}{2} \right\rangle_t + F(x_0)G_{z_0 z_0}(z_0)\left\langle \frac{\delta z^2}{2} \right\rangle_t, \tag{27}$$

recalling that the functions $(\delta x, \delta z)$ are periodic around $(x_0, z_0)$, this means

$$\langle u(X(t), Z(t)) \rangle_t = F(x_0)G(z_0) + F_{x_0 x_0}(x_0)G(z_0)\frac{Var(\delta x)}{2} + F(x_0)G_{z_0 z_0}(z_0)\frac{Var(\delta z)}{2}. \tag{28}$$

Comparing the results obtained in (14) and (28), we identify that, as in the one-dimensional case, the second-order approximation only incorporates the effect of the curvature of the vertical velocity profile in the horizontal direction.

### 2.3. Physical Interpretation: Case A—Wave Orbital Velocities

The above mathematical development can be applied to any velocity field, fulfilling the different cases' hypotheses.

We now turn to its application for measuring near-surface current from a moving buoy. In this section, we focus on the orbital velocities generated by a monochromatic linear wave,

$$\eta = a\cos(kx - \omega t) \tag{29}$$

$$u = \omega\eta \exp(kz) = a\omega \cos(kx - \omega t)\exp(kz). \tag{30}$$

where $u$, can be separate as

$$\begin{aligned} F(x, t) &= a\omega \cos(kx - \omega t) \\ G(z) &= \exp(kz) \end{aligned} \tag{31}$$

Different types of sensor displacements $(\delta x, \delta z)$ with respect to a reference position $(x_0, z_0)$ are considered.

(a)    Case A1: Vertical displacement without vertical variation

We first consider that the sensor has no horizontal displacement ($\delta x = 0$) and that periodic vertical displacement around $z_0$ is given by $\eta$. It corresponds to a current meter attached to a mooring line which would move vertically following its surface float (Figure 3a). From (9), we have:

$$\delta z = \eta \quad \Rightarrow \quad \boxed{\langle u(x, z(t), t) \rangle_t = \frac{1}{2}a^2\omega k \exp(kz_0)} \tag{32}$$

The Equation (32) corresponds to the wave-induced bias obtained by [5,6]. As shown in Figure 3a, the mean current measurement corresponds to half of the Stokes drift at the surface and then exhibits a less pronounced vertical decay than the Stokes drift, which is given by

$$u_s = a^2\omega k e^{2kz_0}. \tag{33}$$

(b)    Case A2: Vertical and horizontal displacement without vertical variation

Now, we consider that $\delta z$ and $\delta x$ correspond to the trajectories described by surface fluid particles in the presence of a monochromatic wave. It corresponds to a current meter attached to a mooring line which would move horizontally and vertically following its surface float. In agreement with linear wave theory in deep water, the float and thus the current meter at depth would describe circular trajectories (Figure 3b). From (23), we have

$$\left. \begin{array}{l} \delta z = \eta = a\cos(kx - \omega t) \\ \delta x = -a\sin(kx - \omega t) \end{array} \right\} \quad \Rightarrow \quad \boxed{\langle u(x(t), z(t), t)\rangle_t = a^2 \omega k e^{kz_0}} \tag{34}$$

Unlike case A1, in the present case, the average velocity measured at the surface will correspond to the Stokes drift. Its vertical decay is less pronounced than the Stokes drift, similar to case A1.

(c)　Case A3: Vertical and horizontal displacement with exponential vertical variation

Here, we consider that the vertical and horizontal variations in a current meter throughout the water column follow the orbital trajectories, with an attenuation concerning the surface as the depth increases, equal to $e^{kz}$ (Figure 3c). This case would correspond to the measurement obtained by a sensor mounted on a buoy, which would perfectly follow the wave orbital velocities at its sampling depth. From (23), we have

$$\left. \begin{array}{l} \delta z = a\cos(kx - \omega t)\exp(kz) \\ \delta x = -a\sin(kx - \omega t)\exp(kz) \end{array} \right\} \quad \Rightarrow \quad \boxed{\langle u(x(t), z(t), t)\rangle_t = a^2 \omega k e^{2kz_0}} \tag{35}$$

Equation (35) agrees with the results of [6]. It indicates that if we consider a set of current meters moving vertically and horizontally, following circular trajectories set by the wave orbital velocities, the wave bias corresponds to the Stokes drift (Figure 3c). In the present case, the Stokes drift is being fully sampled.

(d)　Case A4: Vertical and horizontal displacement with different vertical variations

For this case, we consider a wave-following buoy, to which an array of current meters is attached, free to move horizontally following the orbital trajectories induced by the presence of waves ($\delta x = -a\sin(kx - \omega t)\exp(kz)$), but where the vertical displacement is constrained by the elevation of the sea surface ($\delta z = \eta$). For this case, we obtain the following equation by using (23):

$$\left. \begin{array}{l} \delta z = a\cos(kx - \omega t) \\ \delta x = -a\sin(kx - \omega t)\exp(kz) \end{array} \right\} \quad \Rightarrow \quad \boxed{\langle u(x(t), z(t), t)\rangle_t = \frac{1}{2}a^2 \omega k [e^{2kz_0} + e^{kz_0}]} \tag{36}$$

The trajectory of a current meter located at the surface describes a perfect circle. However, as it moves away from the surface, the circular trajectory becomes an ellipse with the major axis given by the wave amplitude and the minor axis given by the wave-induced orbital trajectories in the horizontal (Figure 3d). The mean current measurements recorded by the current meters will correspond to the Stokes drift at the surface; however, due to the constraint on the vertical displacement, the amplitude of the horizontal displacement decreases as the depth increases, and the vertical displacement will prevail. Therefore, the present case corresponds to a combination of cases A1 and A3.

(e)　Case A5: Measurement made on a buoy with horizontal inclination

In order to obtain the measurement that would be obtained by a current meter attached to a buoy with a particular horizontal inclination, we will consider a variation of case A4. Similarly to previous cases, $\delta z = \eta$, but $\delta x$ below the surface is not influenced by the wave-induced trajectory but is instead restricted to an imposed profile that, for practical purposes, is represented by a linear function (see Figure 3e). We then have

$$\left. \begin{array}{l} \delta z = a\cos(kx - \omega t) \\ \delta x = -a\sin(kx - \omega t)f(z) \end{array} \right\} \quad \Rightarrow \quad \boxed{\langle u(x(t), z(t), t)\rangle_t = \frac{1}{2}a^2 \omega k e^{kz_0}[f(z) + 1]} \tag{37}$$

The trajectory described by a surface current meter corresponds to a perfect circle (similar to cases 2–4), while in the vertical direction, the horizontal trajectories decay

linearly, and a depth of no motion will be present, which depends on the characteristics of the considered function. The mean velocity field sampled by a surface current meter will correspond to the wave Stokes drift. Right at a depth of no motion, the mean velocity recorded by a current meter will be the same as obtained in case A1 (Figure 3e).

The cases presented in this section are summarized in Table 1, with the trajectories of the current meters $\delta x$ and $\delta z$, and the form of the mean current.

**Table 1.** Summary of the different case studies. We consider monochromatic waves. For numerical calculations, we use an illustrative profile $f(z) = 1 + (1/nm)z$, where $nm$ is the depth of no motion set to 20 m.

|     |         | $\delta \mathbf{x}$ | $\delta \mathbf{z}$ | $\langle u \rangle_t$ |
| --- | ------- | ------------------- | ------------------- | --------------------- |
| (a) | Case A1 | $0$ | $a\cos(\mathbf{k}\cdot\mathbf{x} - \omega t)$ | $\frac{1}{2}a^2\omega k \exp\{kz_0\}$ |
| (b) | Case A2 | $-a\sin(\mathbf{k}\cdot\mathbf{x} - \omega t)$ | $a\cos(\mathbf{k}\cdot\mathbf{x} - \omega t)$ | $a^2\omega k \exp\{kz_0\}$ |
| (c) | Case A3 | $-a\sin(\mathbf{k}\cdot\mathbf{x} - \omega t)\exp\{kz\}$ | $a\cos(\mathbf{k}\cdot\mathbf{x} - \omega t)\exp\{kz\}$ | $a^2\omega k \exp\{2kz_0\}$ |
| (d) | Case A4 | $-a\sin(\mathbf{k}\cdot\mathbf{x} - \omega t)\exp\{kz\}$ | $a\cos(\mathbf{k}\cdot\mathbf{x} - \omega t)$ | $\frac{1}{2}a^2\omega k[\exp\{2kz_0\} + \exp\{kz_0\}]$ |
| (e) | Case A5 | $-a\sin(\mathbf{k}\cdot\mathbf{x} - \omega t)f(z)$ | $a\cos(\mathbf{k}\cdot\mathbf{x} - \omega t)$ | $\frac{1}{2}a^2\omega k \exp\{kz_0\}[1 + f(z_0)]$ |

Note that in the monochromatic case studied in this section, a current meter would never rise above the water level and will always remain inside the water, even though the mean measuring depth is above the wave trough (shaded region in Figure 3). If we would consider Eulerian measurements or measurements from buoys of finite dimensions, the problem of a current meter leaving the water would arise, as will be dealt with later on.

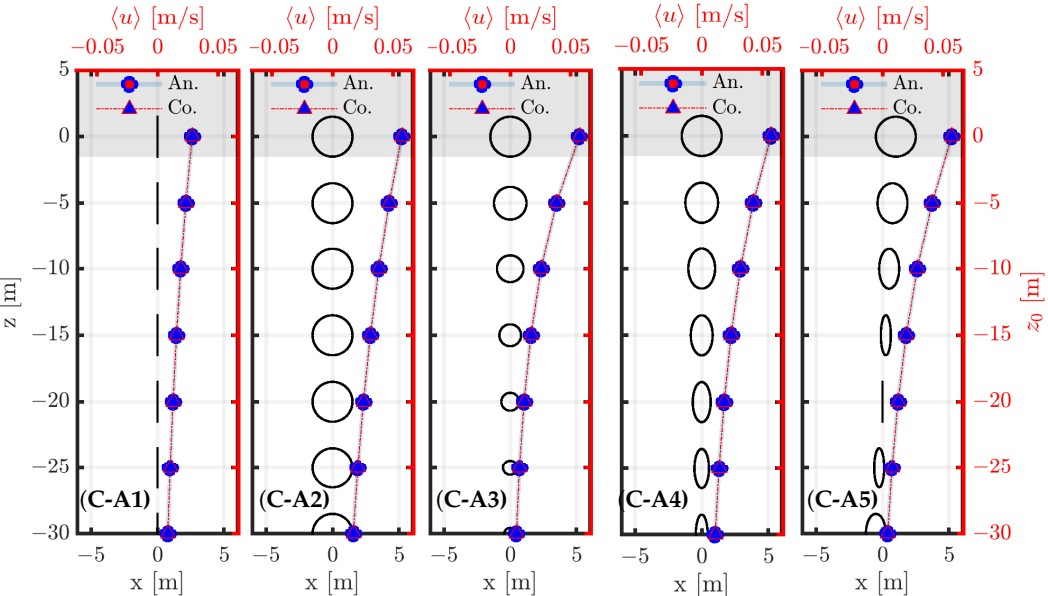

**Figure 3.** Cases A1 to A5 refer to different studies applied to a monochromatic wave. The black line represents the trajectory (in meters) described by a current meter for a given depth and refers to the left (black) axis. The cyan line with blue circles and the red line with red triangles represent the analytical and numerical solutions from a numerical model presented in Section 2.5 obtained by performing a temporal averaging on the velocity field (in m/s) and are referred to the right (red) axis; these are superimposed because both solutions are equal. The shaded region indicates depths above the wave trough. In all cases, a wave amplitude of 1.5 m and a period of 10 s are considered in deep-water conditions.

### 2.4. Physical Interpretation: Case B: Permanent Current with Vertical Shear

In the previous results, we analyzed specific cases of a horizontal velocity field of the type $u = u(x, z, t)$, where only the horizontal component of orbital wave velocities was considered. This component has the characteristic of being in phase with the surface. If we consider horizontal and vertical variations of a current meter associated with the presence of waves, but the velocity field does not have significant horizontal variations or these are much smaller compared to the displacements associated with orbital velocities $(F(x) << \delta x)$, then Equation (28) reduces to Equation (23), and we would obtain the same result as in case 1b.

In this case, we consider a velocity field that is not in phase with the wave, independent of time and coordinates $x$ and $y$, but with a vertical shear $u = u(z)$. If we make measurements with a current meter mounted on a buoy that responds to the wave orbital velocities at the surface, we can obtain from (13):

$$\left.\begin{array}{c} \delta z = a\cos(kx - \omega t) \\ F(x, t) = 1 \\ G(z) = f(z) \end{array}\right\} \quad \Rightarrow \quad \begin{array}{c} \langle u(z(t))\rangle_t = f(z_0) + f''(z_0)\left\langle \frac{\eta^2}{2}\right\rangle_t \\ \\ \boxed{\langle u(z(t))\rangle_t = f(z_0) + \frac{1}{4}a^2 f''(z_0)} \end{array} \tag{38}$$

From Equation (38), we can identify that for this particular case, the difference between performing an Eulerian or undulating average on the velocity field with a vertical shear will increase with the current curvature.

### 2.5. Velocity Field Model

In this section, we wish to evaluate the different wave biases obtained previously in realistic oceanic conditions. For that purpose, the ocean velocity field is constructed by considering a model for a specific wind $u_{10}$ and fetch $\chi$. The velocity field is constructed in a framework that follows the free surface $\eta(t)$ at each instant; this allows us to generate a very fine-resolution mesh near the surface, which is subsequently interpolated to a mesh referred to as a fixed vertical coordinate. The velocity field contains wave orbital velocities computed from a general wind wave spectrum plus an Ekman-type permanent current with vertical shear.

#### 2.5.1. Wind Sea Spectrum

The wave spectrum is calculated in the model following [15] (referred to as DHH from here on), by considering a constant wind and fetch.

$$F(\omega) = \frac{\alpha g^2}{\omega^4 \omega_p} \exp\left[-\frac{5}{4}\left(\frac{\omega_p}{\omega}\right)^4\right] \cdot \gamma^\Gamma \tag{39}$$

where $\Gamma$ corresponds to

$$\Gamma = \exp\left\{-\frac{(\omega - \omega_p)^2}{2\sigma^2 \omega_p^2}\right\}$$

and $\omega_p$, $\alpha$, $\gamma$, and $\sigma$ are the angular peak period of the spectrum, the equilibrium range parameter, the enhancement factor, and the peak width parameter, respectively.

According to [15], the spectral characteristics are related to the fetch and wind intensity as follows:

$$\alpha = 0.006\left(\frac{u_{10}}{c_p}\right)^{0.55}; \quad 0.83 < \frac{u_{10}}{c_p} < 5. \tag{40}$$

$$\sigma = 0.08\left[1 + \frac{4}{\left(\frac{u_{10}}{c_p}\right)^3}\right]; \quad 0.83 < \frac{u_{10}}{c_p} < 5. \tag{41}$$

$$\gamma = \begin{cases} 1.7, & 0.83 < \frac{u_{10}}{c_p} < 1; \\ 1.7 + 6\log\left(\frac{u_{10}}{c_p}\right), & 0.83 < \frac{u_{10}}{c_p} < 1. \end{cases} \tag{42}$$

$$\frac{u_{10}}{c_p} = 11.6\tilde{\chi}^{-0.23}. \tag{43}$$

where $\tilde{\chi}$ corresponds to the non-dimensional fetch, given by $\tilde{\chi} = \frac{\chi g}{u_{10}^2}$, and the angular frequency associated with the spectral peak is given by

$$\omega_p = \frac{2\pi}{T_p} = \frac{2\pi}{0.54 g^{-0.77} u_{10}^{0.54} \chi^{0.23}}. \tag{44}$$

To consider realistic wave conditions, experiments were conducted with a wide range of wind and fetch. Various wave parameters were calculated for each case, including the wave age $c_p/u_{10}$, the frequency associated with the spectral peak $f_p$, the significant wave height $H_s$ and the Stokes drift (Figure 4).

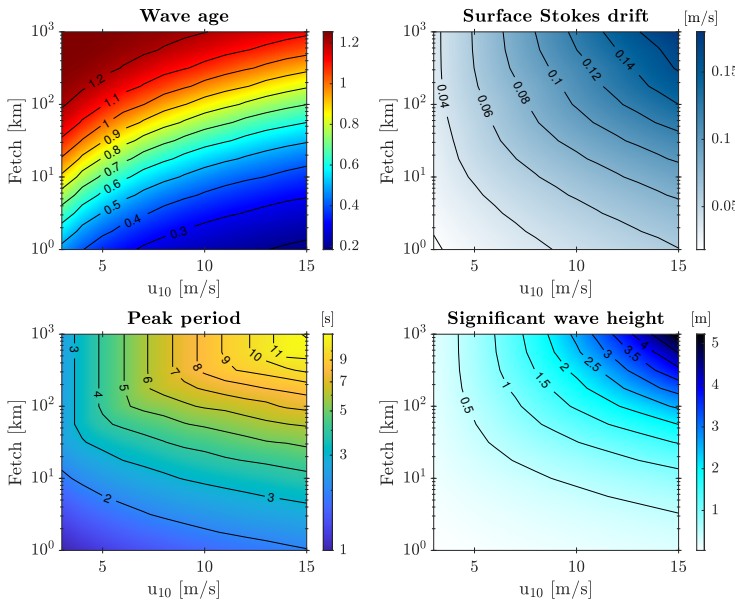

**Figure 4.** State diagrams were plotted as a function of wind speed and fetch for the wave age $c_p/u_{10}$ (**upper left panel**), surface Stokes drift (**upper right panel**), a period associated with the spectral peak (**lower left panel**), and significant wave height (**lower right panel**).

2.5.2. Orbital Velocity Field: Random Phase Model from a Realistic Wind Sea Spectrum

The free surface elevation corresponding to the generated wave spectrum is obtained using a random phase model. For each energy band of the DHH spectrum, the amplitude is calculated, and deep-water conditions are assumed in the wave dispersion relation such that the total free-surface elevation is

$$\eta_{tot}(x,t) = \sum_{i=1}^{N} \eta_i = \sum_{i=1}^{N} a_i \cos(k_i x - \omega_i t - \phi_i), \tag{45}$$

where

$$a(f) = 4\sqrt{S(f)\Delta f} \tag{46}$$

and

$$\omega^2 = gk. \tag{47}$$

The total induced velocity field $u_{tot}$ due to the numerically generated irregular waves is calculated using a modified version of the current model proposed by Donelan et al. (1992) [16]. The model considers the linear superposition of waves that propagate freely, such that shorter waves travel on top of longer ones, and together they determine the total surface elevation. Each incorporated wave satisfies the condition of infinitesimal slopes, so we can apply the corresponding boundary conditions of linear wave theory, such that $u(z = \eta) \rightarrow u(z = 0)$. The modification of the model is due to

the assumption that each wave propagates freely over the surface described by the other waves. Therefore no coupling between short and long waves is considered (there is no modulation transfer function between short and long waves [17]). In addition, the model directly solves the orbital velocity field in a framework that follows the surface rather than solving the system of equations in a system of fixed Eulerian coordinate frameworks as proposed by [16]. By doing this, the generated mesh can include a finer vertical resolution close to the surface, thus reducing processing time.

To achieve this, we consider the orbital velocity of a monochromatic wave, whose domain is given by:

$$u(x,z,t) = \begin{cases} a\omega e^{kz} \cos(kx - \omega t) & z \leq \eta \\ 0 & z > \eta \end{cases} \tag{48}$$

The total waves generated by the random phase model form $\eta_{tot}$. Subsequently, upon considering the superposition model, the following is obtained:

$$u_{tot}(x,z,t) = \sum_{i=1}^{N} \omega_i e^{k_i d_i} \eta_i \tag{49}$$

where $d_i = z - (\eta_{tot} - \eta_i)$.

Because each wave satisfies the condition of infinitesimal slopes and it is required to obtain the contribution of each wave concerning the total field of orbital velocities relative to $\eta_{tot}$, it is convenient to superimpose the orbital velocities of each wave on a regular grid, such that $z = 0$ corresponds to $\eta_{tot}$ at the surface; this allows for an increase in vertical resolution very close to the surface (i.e., $z = 0$). For this purpose, we define $x = x_0 + \delta x(t)$ and $z = z_0 + \delta z(t)$, and solve for the following:

$$u(x_0, z_0, t) = a\omega e^{kz} \cos(kx - \omega t) \tag{50}$$

Finally, to reconstruct the total velocity field $u_{tot}$ (49), corresponding to the field in fixed coordinates, an interpolation is performed in the vertical coordinate of the velocity field, considering $z = z_0 + \eta_{tot}$. To illustrate the velocity field obtained using the proposed model, Figure 5 presents two representations: one in fixed (upper) and the wave-following coordinates (lower) of the reconstructed wave orbital velocities. This figure showcases the first 60 s of a modeled case.

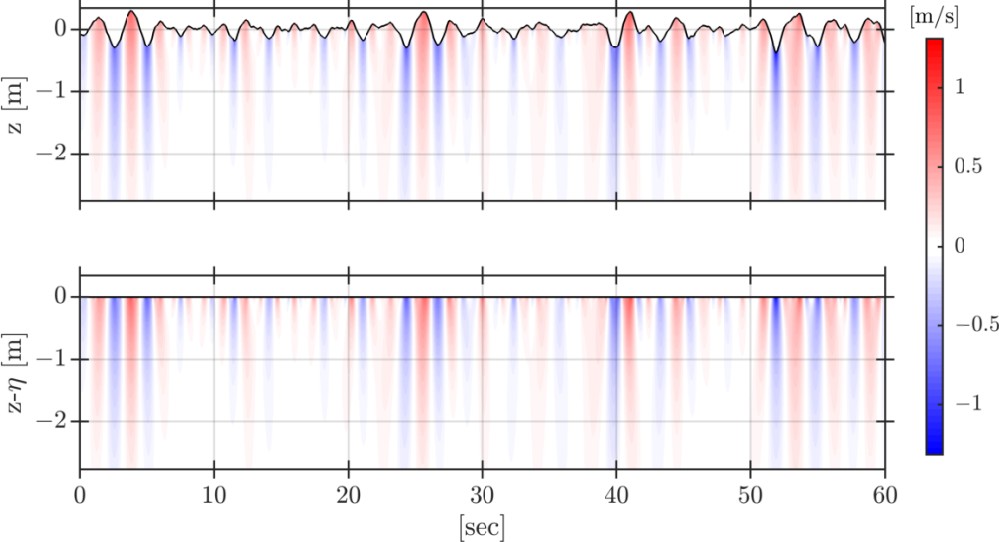

**Figure 5.** Representation of the orbital velocity field based on the proposed model, considering a wind speed of 4 m s$^{-1}$ and a fetch of 32 km. The top and bottom panels correspond to the framework in fixed coordinates and the one following the waves, respectively.

### 2.5.3. Mean Current Field: Realistic Ekman-Type Permanent Current

In addition to the fluctuating wave orbital velocities, an Ekman-type permanent current is incorporated. It is calculated assuming constant wind and constant waves in fetch-limited conditions. For vertical mixing, the unstratified model is used, proposed by [18], where the surface roughness parameter $z_\emptyset$ is parameterized from [19] as $z_\emptyset = 1.6\,H_s$. This velocity field is kept constant in time for the wave-following framework. Examples of near-surface currents are shown in Figure 6.

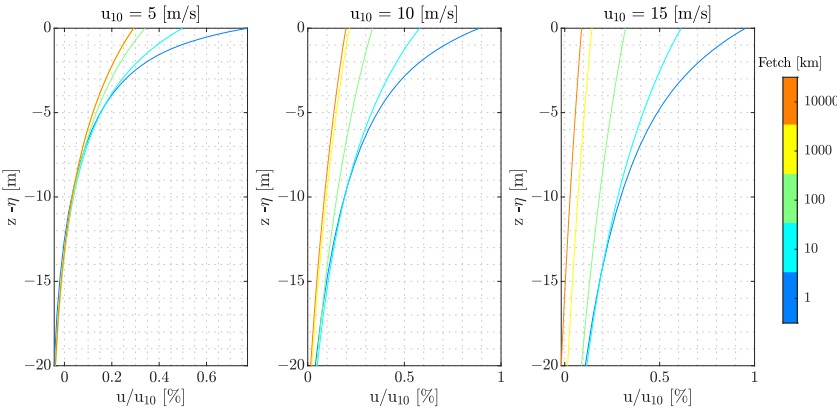

**Figure 6.** Average of the mean velocities (Ekman-type permanent current) in the wave-following framework, normalized by the wind speed $u_{10}$ (m/s) used in the DHH spectrum, expressed in percent. The color bar corresponds to the fetch (km) used in each case.

By incorporating the Ekman-type permanent current in that way, we ensure that its vertical profile corresponds to the average in the undulating framework. We will later consider how this profile is modified when measured in a different reference frame (e.g., in a fixed framework).

Although the dynamics of waves can be affected by the presence of a vertically sheared current [20,21], for simplicity, we consider both velocity fields as independent and therefore ignore this effect.

### 2.5.4. Buoy Motion Response to the Wave Field

As discussed above, the time-average velocity measurement from a buoy-mounted sensor strongly depends on the buoy motion itself. That motion might not perfectly follow the sea surface motion. The dimensions, buoyancy, size, anchoring, and potential marine fouling will determine the buoy's response to the movement induced by a specific range of waves.

A simple description of the buoy's motion for a particular sea state can be obtained by applying a low-pass frequency filter to the DHH spectrum with a determined cutoff frequency $f_{cut}$. The trajectory of the buoy's center of flotation at rest is then reconstructed using a model with the same random phases, and this is an idealized description where the buoy perfectly responds to low-frequency waves and has no motion induced by waves with a frequency higher than $f_{cut}$.

Different buoy responses can be represented for each spectrum by modifying the values of $f_{cut}$. The extreme cases of motion correspond to the frames of reference in fixed coordinates (a buoy that remains still). They are represented by $f_{cut} \sim 0$, and a frame of reference that perfectly follows the surface (its motion completely responds to the present wave) and is represented by $f_{cut} \sim f_{max}$, where $f_{max}$ corresponds to the maximum frequency used to construct the spectrum. However, many cases that depend on the buoy's response to wave motion will be accounted for.

## 3. Results: Time-Average Velocity Profiles

The velocity fields generated in the previous section are now time-averaged. As mentioned, this average can be performed considering different types of sensor motions.

For analysis purposes, the extreme cases of sensor motion will be referred to by the subscripts E and WF for Eulerian and wave-following averages, respectively. Intermediate cases, which correspond to the general case of undulating mean, will be referred to by the subscript $f_{\text{cut}}$, which is the maximum frequency to which the sensor and its mooring line respond. The time average of the horizontal component of the orbital velocity field will be called $U^W$. The time average of the Ekman-type permanent current will be called $U^{Ek}$. In this section, we will focus on describing the error that arises in calculating mean currents when only the horizontal displacements induced by waves on a buoy or mooring line are considered.

### 3.1. Time Average of the Wave Orbital Velocity Field

Let us consider first the average of the wave orbital velocity field, with no Ekman current. In the case of a monochromatic wave, an Eulerian averaging gives a mass transport with a parabolic profile, as discussed above. When considering a more complex irregular sea state, the Stokes transport modifies its parabolic profile since the averaged profile corresponds to the sum of all waves considered in the random phase model with different wave numbers. This is illustrated in Figure 7, where the black line represents $U_E^W$. On the other hand, if we consider a current meter that follows the free surface, we obtain the wave-following average $U_{WF}^W$ shown as a red line in Figure 7. It corresponds to the average of individually resolved wave components of the random phase model. That solution correctly fits the analytical results obtained in case A1 by the Equation (32).

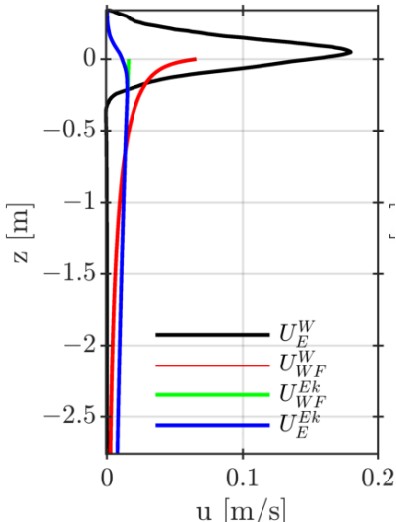

**Figure 7.** Time-average velocity profiles of a wave orbital velocity field of a mean uniform current field. Measurements are considered in two extreme sensor configurations, a motionless sensor (Eulerian mean) and a wave-following sensor. The wind is set to 4 m s$^{-1}$ and the fetch to 32 km.

If we consider the measurement of a sensor mounted on a buoy or mooring line which does not perfectly follow the surface, there will be a lack of measurements when the sensor leaves the water, so the wave pseudo-momentum is not perfectly sampled. The average profile corresponds to a transition between the wave-following and the Eulerian averages, as shown in Figure 8. That figure presents the results of the velocity field averaging for waves generated by a wind of 15 m s$^{-1}$ and a fetch of 100 km, which corresponds to a developed wind sea (wave age $c_p/u_{10} \sim 1$).

In Figure 8a, the black line corresponds to the Eulerian average of the orbital velocity field, and the dotted black line corresponds to the wave-following average, i.e., the average that would be obtained by a sensor mounted on a mooring line, which buoy would move vertically and perfectly follow the free surface. The colored lines correspond to intermediate cases, where the sensor only moves with low-frequency waves with $f < f_{\text{cut}}$.

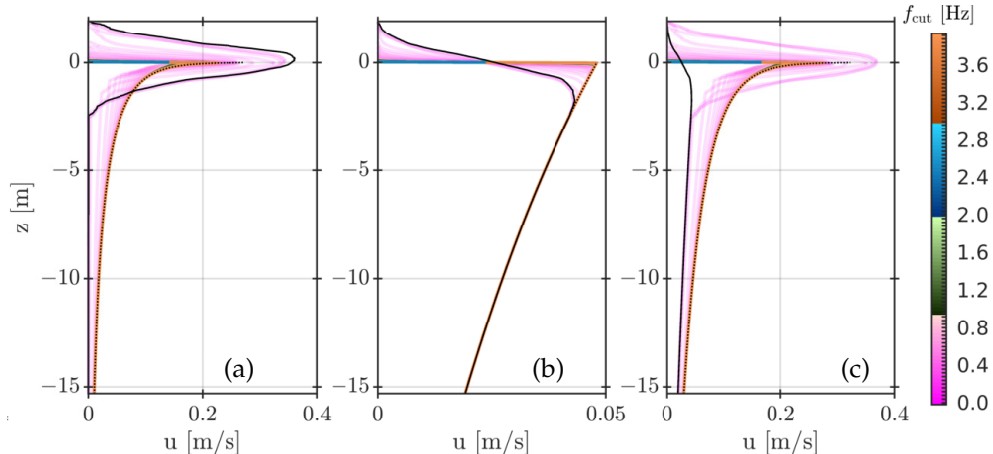

**Figure 8.** Velocity profiles of the time average current fields. (**a**) Orbital velocity field $U^W$. (**b**) Ekman-type permanent current field $U^{Ek}$. (**c**) Total current $U^{Ek} + U^W$. In each plot, the dashed black line represents the wave-following average, while the black line represents the Eulerian average. The color range corresponds to intermediate cases, with the sensor motion following the low frequency ($f < f_{\text{cut}}$) waves only. The wind is set to 15 m s$^{-1}$ and the fetch to 100 km.

*3.2. Time Average of Ekman Current Field*

Let us now consider the Ekman-type permanent current field, considering wave-induced sensor displacements. We will consider an imaginary case with no wave orbital velocity presented to focus on Ekman's current contribution.

In Figure 7, the blue line represents the Eulerian average of the Ekman current, while the green line corresponds to the wave-following average. The main difference is observed between crests and troughs, whereas they tend to be similar below the troughs (typically for $z < -H_s$); this is because the current profiles do not have a curvature large enough to generate a second-order effect in the temporal average, as indicated in hypothesis B in Equation (14). Therefore, below the troughs, the undulating average approximates the Eulerian average.

By observing Figures 4 and 6, it was identified that in the cases with lower fetch and higher wind speed, the mean current profile presents more intense velocities near the surface and has a different curvature. To identify the error that causes a curvature in the mean current, we will examine two wave regimes, namely the young and developing wave cases (see Figures 9 and 10, respectively). Regardless of the type of wave that induces motion on a buoy or mooring line where a current meter is installed, the mean current measurements in the wave framework exhibit no significant error compared to those obtained in an Eulerian framework.

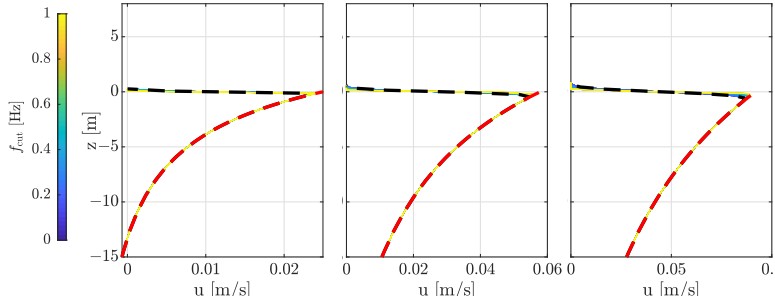

**Figure 9.** Time average profiles of Ekman current for young sea states (fetch of 10 km, wind from left to right $u_{10} = 5, 10, 15$ [m/s]). The Eulerian average is the dashed black line, and the wave-following average is the dashed red line. The color bar indicates the cutoff frequency in the wave-induced sensor motion.

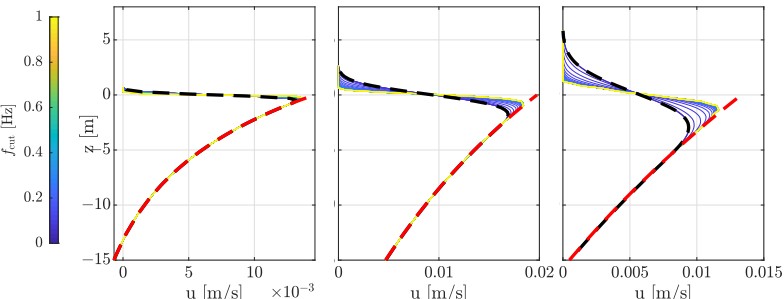

**Figure 10.** The same as the previous figure but for developed sea states (fetch of 10,000 km).

### 3.3. Time Average of the Total Current

The total current field corresponding to the sum of $U_{Ek}$ and $U^W$ was analyzed in order to identify how the average velocity measured by a sensor attached to a buoy or its mooring line is modified considering the wave-induced motion. In Figure 8c, the case is shown where the wind is set to 15 m s$^{-1}$ and the fetch to 100 km. The red line corresponds to $U_{SF}^W + U_{SF}^{Ek}$ and allows us to determine how the average of the measurements would be modified in each case, and the black dashed line represents $U_E^{Ek}$ as a reference. The error obtained below the troughs will depend solely on the mean orbital velocities. When the wave-induced motion on a buoy or its mooring line, where current meters are attached, corresponds to waves with frequencies above 1 Hz, the errors obtained in the average current will be of the same magnitude as the Ekman current.

## 4. Discussion: Error in the Time Average Velocity Measurement

In the previous section, we identified that the error in surface velocity measurement depends on the state of wave development and the magnitude of the Ekman current in comparison to the average orbital velocities. Therefore, a more detailed analysis will be carried out in this section.

### 4.1. Error as a Function of the Sea State

In general, there is a direct relationship between the wave momentum and the wave age, as well as an inverse relationship between the magnitude of the Ekman current and the wave age (Figure 11). However, other factors must be taken into account since the same wave age can occur for different ranges of fetch and wind (Figure 4). Therefore, a more thorough analysis is necessary. As the theoretically obtained results in Section 2.2 indicate, at the surface, the bias in measuring orbital velocities is proportional to the Stokes drift; this value was compared with the wave age and the wind and fetch conditions under which these cases occur (Figure 11).

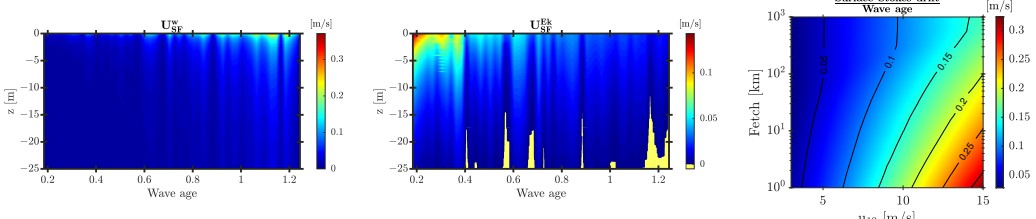

**Figure 11.** Vertical profile of the average wave momentum (**left**) and Ekman current (**center**) with respect to wave age. (**Right**) State diagram of the ratio between surface Stokes drift and wave age as a function of wind [m] and fetch [km].

### 4.2. Error as a Function of the Mooring Line Motion Type

Not only will the characteristics of the mean velocity profile be determined by the wave-induced motion on a buoy and its mooring, but the magnitude of the error obtained from the measurements will also be related to the energy and the wave development

conditions. The magnitude of the Stokes drift is mainly determined by the high-frequency region of the wave spectrum, which, in turn, is influenced by both fetch and wind conditions (see Figure 4).

Due to the various variables that define the profile of the Stokes drift and the Ekman current, it is complex to define a general behavior regarding the characteristics that generate such velocity fields. In order to account for this variety of variables, we defined a new characteristic frequency $f_{3p}$ associated with the peak frequency of the third moment of the spectrum $E(f)f^3$. The third moment of the spectrum is proportional to the Stokes drift in the unidirectional case considered in this work (see Equation (33)). Therefore, $f_{3p}$ represents the frequency of the waves contributing most to the Stokes drift and is significantly higher than the peak frequency $f_p$ which represents waves contributing most to the significant wave height $H_s$.

Figures 12 and 13 show the average velocity profiles obtained for young wave fields and a fully developed wave field, respectively. As in the previous section, the orbital velocity field and the Ekman current field are considered separately (top and bottom row, respectively). The columns show the five different cases of mooring line motion, as described in Section 2.3 (vertically and/or horizontally and/or obliquely moving mooring lines). In each plot, the colors represent the maximum wave frequency $f_{cut}$ to which sensor motion responds, from Eulerian to wave-following. Note that the frequency $f_{cut}$ has been normalized by the frequency associated with the spectral peak of the Stokes drift $f_{3p}$.

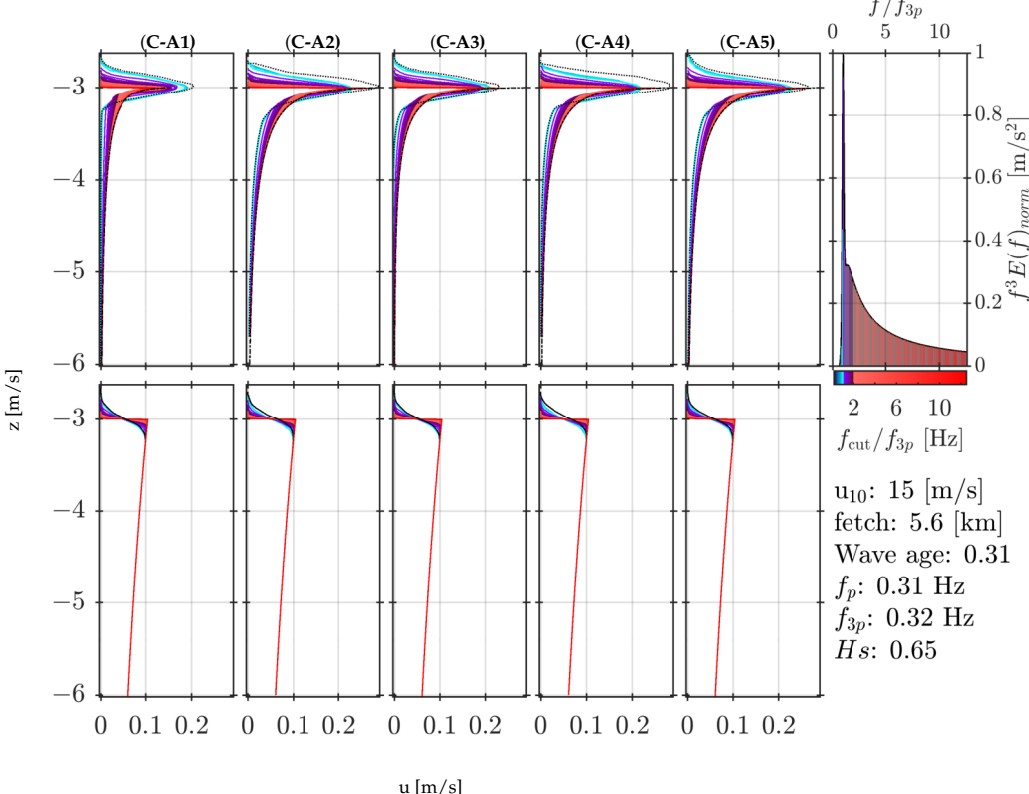

**Figure 12.** Profiles of time averages of wave orbital velocities (**top row**) and of Ekman current (**bottom row**). The columns represent the five different cases of mooring line motions described in Section 2.3. For each plot, different sensor motions are considered, from Eulerian to wave-following, by varying the maximum wave frequency $f_{cut}$ to which the sensor motion responds. The right panel shows the third moment of the frequency spectrum $f^3E(f)$, and the color bar represents the frequency $f_{cut}$ normalized by the frequency associated with the spectral peak of the Stokes drift $f_{3p}$. The wind is set to 15 m s$^{-1}$ and the fetch to 5.6 km, which produces a young sea state (wave age $c_p/u_{10} = 0.31$).

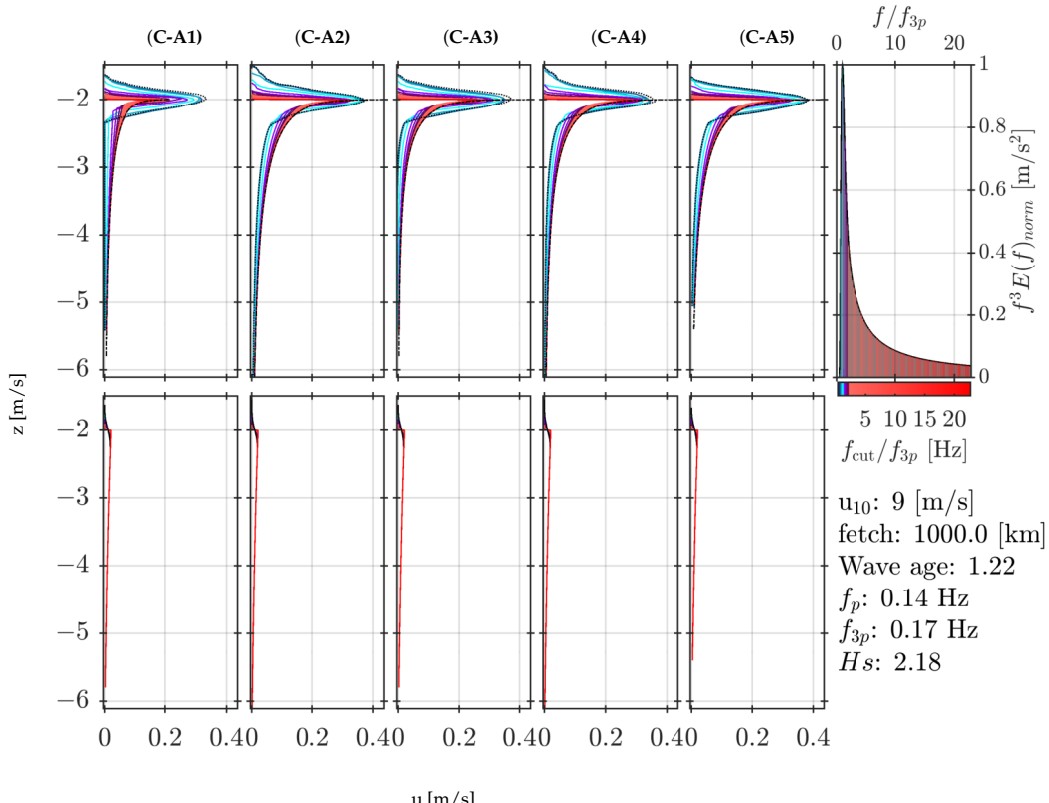

**Figure 13.** The same as the previous figure but for fully-developed waves. The wind is set to 9 m s$^{-1}$ and the fetch to 1000 km, which produces a fully-developed sea state (wave age $c_p/u_{10} = 1.22$).

Whatever the type of mooring line motion (vertical and/or horizontal and/or oblique displacements), it is identified that the normalized frequency $f_{cut}/f_{3p}$ captures the general behavior of the time average, namely:

$$
\begin{array}{c|l}
f_{cut}/f_{3p} \leq 1 & \text{Tends towards the Eulerian mean.} \\
1 < f_{cut}/f_{3p} \leq 2 & \text{Transition region.} \\
2 < f_{cut}/f_{3p} & \text{Tends towards the undulating mean.}
\end{array}
$$

Regarding the magnitude of the velocity averages for the case of an Ekman current, it is identified that the differences are mainly observed in the region confined between $\pm H_s$, as expected due to the absence of measurements. Therefore, measurements made in this region will be most affected by the type of sensor motion, but this region cannot be considered.

In recent years, the precision in measuring surface velocities by different types of current meters (ADCP, ADV) has increased considerably, with a minimum precision in the range of 2–5 mm/s [22]. Regarding the velocity field associated with wave orbital velocities, considering the movement of a current meter presents differences that can be observed in the measurements. It is impossible to mention a depth at which the error in velocity measurement is measurable or not measurable in general since this will depend on various factors, as mentioned in this section.

### 4.3. Quantification of the Error

As observed in Figures 12 and 13, the time average velocity strongly depends on the type of mooring line motion response to waves. In the idealized cases that consider a monochromatic wave, it was mentioned that the Eulerian mean of the orbital velocities is confined between the crests and troughs of the wave (cases 1 and 3 of Section 2.3). It is important to mention that the variations in the position of the current meter (cases 2, 4, and 5) induce an increase in the average velocity. This result is produced because the information

from surface trajectories affects the measurements throughout the water column. We obtain the full Stokes drift profile when considering a system that perfectly follows the free surface, as in case 3.

To quantify the error in the time average velocity measurement, we consider the relative difference between the Stokes drift and the time average of orbital velocity field $U_{f_{cut}}^{W}$ and define the relative error as

$$U_{f_{cut}}^{rel}(z) = \frac{U_{f_{cut}}^{w}(z) - U_s(z)}{U_s(z)} * 100. \tag{51}$$

For the different types of sensor and mooring motions analyzed in this study, we should focus on the region below the trough but at a depth where the magnitude of the Stokes drift is significant. Steer clear of the deeps where the Stokes drift tends to zero, as this could result in unreliable or invalid data. Therefore, we will consider cases where the *e*-folding depth of the Stokes drift $D_s$ is relevant, with

$$D_s = -\frac{1}{2}\frac{1}{\bar{k}} \tag{52}$$

where $\bar{k}$ corresponds to the inverse of a depth scale, in this case, taken as the wave number associated with the spectral peak. As the wave development increases, measurements closer to the surface must be discarded.

The relative error $U_{f_{cut}}^{rel}(z)$ indicates that the average profile obtained by considering each range of movement and the maximum frequency of the waves to which a buoy or a mooring line responds differs by a certain percentage from the Stokes drift at a given depth. A value of $U_{f_{cut}}^{rel} = 0$ indicates that the measurement being taken corresponds to the Stokes drift at a given depth, while $U_{f_{cut}}^{rel} \sim -100$ corresponds to a current measurement where the magnitude is close to zero.

To analyze the relative error to the Stokes drift, we consider the wave ages described previously (Figures 12 and 13). Figures 14 and 15 show the corresponding errors for young and fully-developed waves, respectively. In all the cases considered, when we consider the wave-following average in case A3, we are completely capturing the Stokes drift. Regardless of the degree of wave development, the type of motion that a buoy or mooring line has will determine how we capture the Stokes drift for a given depth; however, the waves' development determines the value being captured.

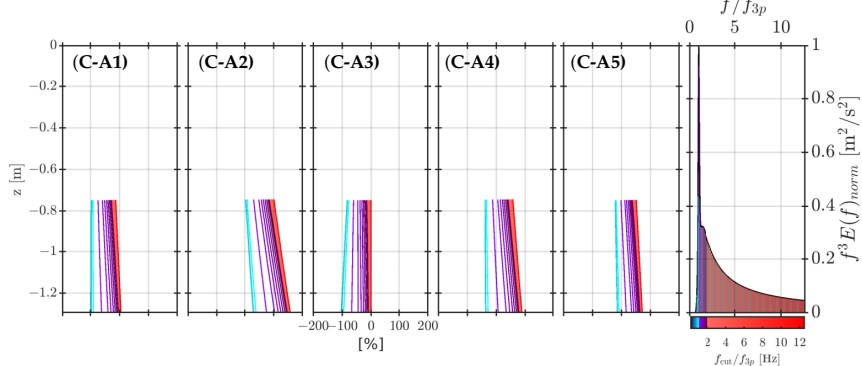

**Figure 14.** Profiles of measurement error $U_{f_{cut}}^{rel}(z)$ relative to the Stokes drift. The columns represent the five different cases of mooring line motions described in Section 2.3. For each plot, different sensor motions are considered, from Eulerian to wave-following, by varying the maximum wave frequency $f_{cut}$ to which the sensor motion responds. The right panel shows the third moment of the frequency spectrum $f^3 E(f)$, and the color bar represents the frequency $f_{cut}$ normalized by the frequency associated with the spectral peak of the Stokes drift $f_{3p}$. The conditions correspond to those of Figure 12, i.e., for a young sea state.

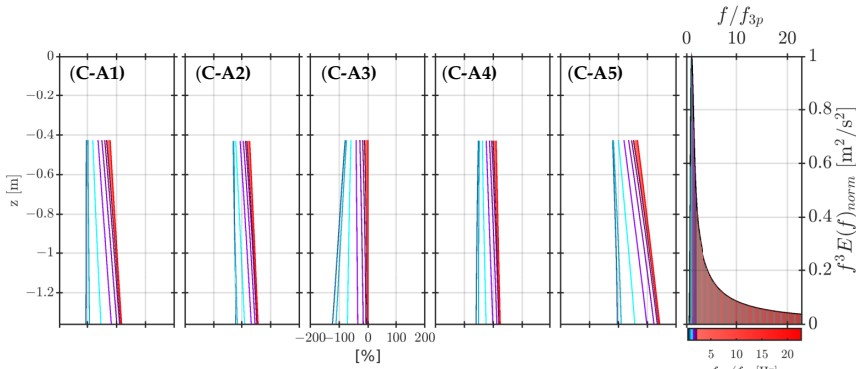

**Figure 15.** The same as the previous figure, but for the conditions of Figure 13, i.e., for fully-developed waves.

From Equations (12) and (13), we can identify that when analyzing $U^{rel}_{f_{cut}}(z)$ concerning the Stokes drift e-folding depth, there is a similar distribution of error regardless of the degree of wave development, and this varies mainly by the type of wave-induced motion on a buoy or mooring line. As the error will be relevant at depths below the wave troughs up to the Stokes drift e-folding depth, $U^{rel}_{f_{cut}}(z)$ will be relevant in conditions of developed waves and energetic waves.

Under these conditions, where the surface Stokes drift can reach values above 0.1 m/s, it should be noted that for certain mooring line motions, the measurement error can exceed the value of the Stokes drift at the sensor depth.

## 5. Conclusions

The measurement of near-surface current from a sensor moving with the waves has been investigated. The sensor motion induces a non-uniform sampling in space and time of the near-surface current field, which can lead to differences or 'errors' once the time average is performed.

We considered two components of the current field—the wave orbital velocities plus an Ekman-type permanent current. Different types of sensor motion or mooring line motion have been considered, including vertical and horizontal motions. The finite size of the sensor or of the buoy, which could lead to a filtering of the response to the short waves, has also been considered. Simple cases with monochromatic waves have been investigated analytically, whereas more realistic cases have been investigated numerically.

It has been identified that the difference between measurements following the waves perfectly, at a fixed location, and from a buoy that responds to low-frequency waves can be considerable under certain conditions. This depends on the relationship between the wave development, the magnitude of velocities near the surface, the vertical decay of its average profile, as well as the sensor motion.

The mean below wave-troughs of an Ekman-type permanent current measured on a buoy or mooring line, influenced by wave-induced motions, regardless of the specific wave motion or response, do not require the consideration of any wave-induced errors. However, the wave orbital velocities produce an error in the average current that can have a similar magnitude to the Ekman-type permanent current; this is especially true when a buoy or mooring line responds to waves with frequencies higher than 1 Hz.

We identify that, linking the wave-induced motions on a buoy or mooring line where a current meter is installed, with respect to the peak frequency of the third moment of the spectrum allows us to classify the mean surface current into three ranges: (1) Eulerian average, (2) wave-following average, and (3) intermediate case of undulating average.

When measurements of surface currents are obtained in the presence of waves, it is important to analyze the depth of the sensor relative to the Stokes drift e-folding depth. This analysis will enable us to determine if the error that occurs in the mean current is significant in relation to the Stokes drift. From this analysis, we can determine the significance of

errors in mean current about Stokes drift by examining the type of wave-induced motion on a buoy or mooring line. Those results can aid in interpreting in situ near-surface current measurements from various devices.

**Author Contributions:** Conceptualization, N.R., F.J.O.-T., P.O. and H.G.-N.; Methodology, C.F.H.-V., N.R., F.J.O.-T., P.O. and H.G.-N.; Investigation, C.F.H.-V.; Writing—original draft, C.F.H.-V.; Writing—review & editing, N.R. and F.J.O.-T.; Visualization, C.F.H.-V.; Supervision, N.R. All authors have read and agreed to the published version of the manuscript.

**Funding:** This research was funded by the National Council of Science and Technology (CONACYT) grant No. 610869.

**Institutional Review Board Statement:** Not applicable.

**Informed Consent Statement:** Not applicable.

**Data Availability Statement:** Not applicable.

**Acknowledgments:** This research has been possible with funds from CONACYT Ph.D. Grant 610869 for C. F. Herrera-Vázquez, and this is a contribution of The Waves Group. We acknowledge the support from CICESE Physical Oceanography Department Graduate Program. Sincere thanks to Julio Sheinbaum and Luis Zavala, for their helpful comments and support. We greatly acknowledge the support and BOMM data availability from CIGoM (SENER-CONACYT project 201441. This work has been funded by Frontiers of Science CONACYT project CF-2019-116328, and by an IFREMER-ESA contract to CICESE. The authors are grateful to the anonymous referees for their constructive comments and suggestions.

**Conflicts of Interest:** The authors declare no conflict of interest.

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
