# Peer review of "On the Measurement of Ocean Near-Surface Current from a Moving Buoy"

_jmse, doi:10.3390/jmse11081534_

Round 1

Reviewer 1 Report

On the Measurement of Ocean Near-Surface Current from a Moving Buoy
by Herrera-Vázquez et al.

The authors present a theoretical analysis of the potential errors induced on current measurements due to the motion of the sensor induced by surface waves. They build on previous studies by considering different configurations accounting for horizontal and vertical displacements, and vertical attenuation. The analysis is carried out for monochromatic waves and broadband spectra with and without a background Ekman current. Overall, I think the manuscript should be of interest to the community and could help with the interpretation of field measurements but requires revisions (mostly editorial). Below I provide a list with specific comments and suggestions.

Specific comments

32-34. Provide references after "... buoys"

49-53. Some of this was already mentioned in lines 37-39. The second sentence seems to be a result of the paper. I suggest removing this short paragraph.

Figure 1a. Why are the vertical velocities large near the crest? It would also be helpful to state which way the wave is going.

99. will give rise --> corresponds

110-112. It is not clear what do you mean by "the current will be modified"? Are you referring to feedbacks?

177. Here \alpha must have units of 1/s if G(z) is dimensionless according to (5).

Equations 24 and 25. G(x) --> G(z0)

Section 2.2. The averaging operator on the left hand side of the equations was dropped.

Figure 3. I don't see cyan lines.

385. The symbol z_0 was used earlier for the mean position. I suggest using different notation for the roughness length to avoid confusion. The value of z0= 1.6 Hs is very large compared to that used by other studies (Terray et al. 1999, Burchard 2001, Moghimi et al. 2016).

Figure 6 is too small. Should the ylabel be z-\eta?

443. 8b--> 8a

453-457. Is the lack of curvature related to the large value of z0 used (i.e., z0= 1.6 Hs)? How sensitive is this result to changes to the value of z0?

460. What do you mean by "as a different curvature"?

Figure 8. I don't see any green lines. I suggest optimizing the color bar/colormap.

Figures 12-15, right panels showing the compensated spectra. The color bar would be better with a logarithmic scale for the fcut/f3p axis.

498-500. Redundant consider revising.

Equation (54). Why not use a Stokes depth based on the wavenumber corresponding to f3p instead of the spectral peak wavenumber? Or an integral measure of the stokes drift depth for example equation 7 in Sullivan et al. 2012.
Section 4.3 argues that the errors are relevant for the region below the crests and above the e-folding depth based on the peak wavenumber. However, from the analysis is unclear if that is the case because the relative errors tend to increase with increasing depth (away from the surface). In other words, is the e-folding depth as a measure of where the errors may be large too conservative?

562. (c) --> A3 (?)

591-593. Not very clear. Consider revising. The same applies to 597-600.

608. delete 'Referencias'

References

Burchard H (2001) Simulating the wave-enchanced layer under breaking surface waves with two-equation turbulence models. Journal of Physical Oceanography, 31(11):3133–3145.

Terray EA, Drennan WM, Donelan MA (1999) The Vertical Structure of Shear and Dissipation in the Ocean Surface Layer. ASI Conference Proceedings - Sydney

Moghimi S, Thomson J, Özkan-Haller T, Umlauf L, Zippel S (2016) On the modeling of wave-enhanced turbulence nearshore. Ocean Modelling, 103:118–132. https://doi.org/10.1016/j.ocemod.2015.11.004

Sullivan PP, Romero L, McWilliams JC, Melville WK (2012) Transient Evolution of Langmuir Turbulence in Ocean Boundary Layers Driven by Hurricane Winds and Waves. Journal of Physical Oceanography, 42(11):1959–1980. https://doi.org/10.1175/JPO-D-12-025.1

line 6. "with a" --> "with"

7. "surface.wave-orbital" --> "surface. Wave orbital"

17. 'indeed' --> "most" (?)

27. delete "of some"

29. "ocean and atmosphere" --> the ocean and the atmosphere

47. "is pending" --> "has not been carried out"

65. "anchored" --> moored. "installed" --> "mounted"

70-71. I suggest replacing "Firstly, an overview of the frameworks used to describe measurements made from moving buoys is present" with "First, an overview of the framework used to describe measurements made from moving buoys is presented"

Section 2.1 and throughout the document - the structuring of the paragraphs could be improved

70 (and other instances). wave-orbital --> wave orbital

133. proposed --> introduced

196. know --> now

199. 'a velocity' --> the velocity, 'is given as' --> is given by

252. ow --> now

264. Do you mean a subsurface buoy instead of an AUV?

332. correspond to --> are the

Author Response

Thank you very much for your valuable comments and feedback.

Reviewer 2 Report

Please see enclosed pdf file.  This manuscript has many errors that need to be corrected before a serious consideration for publication can be given

Most important objection:

On top of page 8, in lines 212-213, you make the mistake of confusing the small first-order quantities (delta x, delta z) with the finite zeroth-order quantities (sin theta, cos theta, 1).  For this reason you end up with the totally incorrect conclusion that the tiny third-order quantity delta z^2 delta z should be represented by the much larger first-order quantity delta z.

I am afraid that this elementary confusion of regular perturbation theory possibly renders the rest of you manuscript totally flawed?

Author Response

Many thanks for your remarks and comments. All of them have been taken into account.

Reviewer 3 Report

Review of "On the Measurement of Ocean Near-Surface Current from a Moving Buoy"

Summary:
The aim of the paper is to investigate the challenges associated with accurately measuring ocean surface currents using in situ measurements on a moving buoy. The authors focus on the biases introduced by wave-induced motion in traditional current meters installed on buoys and aim to investigate the errors in current measurements when a current meter is mounted on a buoy that responds to specific wave scales. The main strengths of the paper lie in its comprehensive overview of the topic and its numerical investigation of current measurement errors.

Specific Comments:
While English is not my native language, I refrain from evaluating the overall quality of the English in this paper. However, I noticed a few sentences with typographical errors or room for improvement in their structure. Therefore, I recommend conducting a thorough review of the text.

The majority of the references cited in the paper are older than five years. It would be advantageous to include an updated literature review on this topic to ensure the incorporation of the latest research findings.

Author Response

Thank you very much for your comments; we have considered them and worked to fulfill them. This resulted in a much better-quality document.
